# Effects of a Football Simulated Exercise on Injury Risk Factors for Anterior Cruciate Ligament (ACL) Injury in Amateur Female Players

**DOI:** 10.3390/biology12010124

**Published:** 2023-01-12

**Authors:** Harriet Ferguson, Jessica Piquet, Monèm Jemni, Anne Delextrat

**Affiliations:** 1Department of Sport and Health Sciences and Social Work, Oxford Brookes University, Oxford OX3 8HU, UK; 2The Carrick Institute of Neuroscience, Cap Canaveral, FL 32920, USA

**Keywords:** fatigue, neuromuscular, angle specific H/Q, rate of torque development

## Abstract

**Simple Summary:**

Women are more at risk of knee injury affecting the Anterior Cruciate Ligament (ACL) compared to men. However, there is limited literature on risk factors linked to strength in female footballers. The aim of this study was to investigate the effects of fatigue on these risk factors. Thirty-three amateur players (20.3 ± 2.0 years old, 1.67 ± 9.31 m, 63.4 ±8.1 kg, 23.6 ± 5.7% body fat) performed strength assessments of the quadriceps and hamstrings on both legs on an isokinetic dynamometer, before and immediately after a football-specific exercise. Results showed fatigue significantly decreased peak knee flexor strength, the ratio of strength between muscles and the speed at which knee flexor strength is produced (−8.8 to −17.0%) in both legs Furthermore, significant decreases in the ratio of strength between muscles were observed at 10° only in the dominant leg (−15.5%), and at 10°, 20° and 30° in the non-dominant leg (−15.1 to −21.8%). These results suggest a reduced capacity of the hamstrings to stabilise the knee joint with fatigue. Unlike results previously shown on men, the non-dominant leg seemed more affected, highlighting the need to consider specific prevention measures in females.

**Abstract:**

Females are more at risk of Anterior Cruciate Ligament (ACL) injuries than males; however, there is limited literature on neuromuscular risk factors such as angle-specific hamstring/quadriceps functional strength ratios (H_ecc_/Q_con_) and rate of torque development (RTD) in female footballers. The aim of this study was to investigate the effects of fatigue on these neuromuscular risk factors. Thirty-three amateur players (20.3 ± 2.0 years old, 1.67 ± 9.31 m, 63.4 ±8.1 kg, 23.6 ± 5.7% body fat) performed strength assessments of the quadriceps (concentrically, Q_con_) and hamstrings (eccentrically, H_ecc_) on both legs on an isokinetic dynamometer, before and immediately after a football-specific exercise. Results showed significantly lower peak H_ecc_ (−15.1 to −15.5%), peak H_ecc_/Q_con_ (−8.8 to −12.9%) and RTD (−14.0 to −17.0%) for hamstring eccentric contractions after fatigue in the dominant and non-dominant legs. Furthermore, significant decreases in H_ecc_/Q_con_ were observed at 10° only in the dominant leg (−15.5%), and at 10°, 20° and 30° in the non-dominant leg (−15.1 to −21.8%). These results suggest a reduced capacity of the hamstrings to stabilise the knee joint with fatigue. Unlike results previously shown on men, the non-dominant leg seemed more affected, highlighting the need to consider specific prevention measures in females.

## 1. Introduction

Football is getting increasingly popular amongst females, as shown by the rise in the number of affiliated teams in the past few years and the 29 million girls and women currently playing football worldwide (fifa.com). This growing interest has increased the physical demands on the players, leading to more injuries (27.6 injury incidences/1000 h exposure in international female footballers [1]). Interestingly, this survey also revealed a greater burden of injuries in females than males (506.7 vs. 454.0 days of absence per 1000 h, accordingly [1]). This could be partially accounted for by the fact that compared to males, female football players are three times more likely to sustain an anterior cruciate ligament (ACL) injury [2]. ACL injury is not only characterised by a frequent need for surgery (hence high burden for players and teams), but also a very high and increasing economic cost [3]. While it is well established that some ACL injuries can be prevented [4], more work is needed to understand injury patterns and modifiable risk factors in female football players, as the majority of publications in this area concern males [1]. 

Neuromuscular risk factors for ACL injuries are commonly cited in the literature as essential modifiable factors [5]. Several prospective studies on male and female athletes have shown that low peak concentric (H_con_) or eccentric hamstring strength (H_ecc_), in particular relative to the concentric strength of the quadriceps (Q_con_), was associated with a greater incidence of ACL injuries later in the season [5,6]. This crucial role of the hamstring could be explained by its involvement in decelerating the forward translation of the tibia in relation to the femur during powerful quadriceps contractions, hence acting as an ACL agonist. However, most studies that have calculated the conventional (H_con_/Q_con_) or functional (H_ecc_/Q_con_) H/Q ratio have used the peak torque values, which usually occur mid-range [7], while it is well established that ACL injuries occur close to knee extension [8]). A better characterisation of players’ injury risk should involve looking at the entire torque–angle relation [7], or specifically at angles close to extension [9]. In addition to the level of force produced by the hamstring, it appears that its capacity to quickly produce force (rate of force or torque development) is a strong predictor of ACL injuries in female elite team sport players [10]. 

Fatigue is another risk factor for ACL injuries frequently cited in the literature [11]. In competition, it is illustrated by the greater number of injuries occurring towards the end of matches [12]. While the effects of fatigue induced by football on neuromuscular risk factors for ACL injuries have been studied in males by many authors, there is a lack of similar studies in females (one out of 10 studies in the systematic review by Silva et al. [13]). Findings from male footballers cannot be applied to female players, because gender-specific metabolic and muscular properties (greater oxidative capacity, mitochondrial intrinsic respiratory rates, capillary density per unit of skeletal muscle and a larger proportion of less fatigable type I muscle fibres in females) suggest that females might be more resistant to fatigue than males [14,15]. 

The few studies that investigated the effects of football-induced fatigue on neuromuscular risk factors for ACL injuries in female players showed significant decreases in H_ecc_ (−8.4 to −17.5%) and H_ecc_/Q_con_ (−8.0 to −14.1%) following simulated or real football matches [9,11,16]. In addition, de Ste Croix et al. [17] measured a significantly longer electromechanical delay (+58.4%) during eccentric hamstring contractions after a simulated football game in elite female youth players, indicating a reduced capacity to quickly contract these muscles. The same team showed that following a simulated football game in elite youth female players, H_ecc_/Q_con_ was significantly reduced (absolute change of −0.27) between 0 and 10° from full knee extension [9]. This highlights a reduced capacity of the hamstring to stabilise the knee joint close to extension. While these results are interesting, they were obtained on females of various ages and playing levels, and to our knowledge, there is no study investigating all risk factors previously mentioned within the same settings.

Therefore, the aim of the present study was to investigate the effects of a simulated football exercise and the angle of knee flexion on a range of risk factors for ACL injuries in female players. A secondary aim was to explore whether the change in these risk factors was related to physiological parameters and performance during the simulated football exercise.

## 2. Materials and Methods

### 2.1. Participants

Thirty-three female football players volunteered to take part (20.3 ± 2.0 years old, 1.67 ± 9.31 m, 63.4 ±8.1 kg, 23.6 ± 5.7% body fat). An a priori sample size determination (G*power v3) using an effect size of 0.5, α of 0.05, β of 0.2 and power of 0.8 revealed a total number of participants of 40 for one of the main risk factors considered. Therefore, our sample size was a bit small; however, the power analysis for the other risk factors showed smaller sample sizes required (n = 26), and hence, we believe that, overall, the number of participants tested does not affect the validity of our findings. They were recruited from a university team and one local football club competing at similar amateur levels. After getting coaches’ agreement, a member of the research team attended a practice session to explain the details of the study to players, and those interested subsequently signed up for it by email. At the time of the study, the teams were involved in two football practice sessions and one match weekly, with no specific strength and conditioning sessions. Exclusion criteria included: not being an outfield player, any lower limb injury in the past 6 months, or an ACL or hamstring injury in the past 2 years. Participants gave written informed consent and the study was approved by the local University ethics committee (reference number 120688) in accordance with the principles set in the Helsinki declaration. 

### 2.2. Procedures

All participants attended the laboratory on two occasions: the first visit was to measure some anthropometric data, and to get familiar with the isokinetic dynamometer (running through the entire protocol) and the simulated football test (two laps). On the second visit, participants’ maximal strength for the quadriceps and hamstrings were tested on an isokinetic dynamometer before and immediately after (maximum time of 5 min to walk from the pitch to the lab) they took part in a 90-min simulated football exercise. Participants were instructed to refrain from consuming alcohol and caffeine 24 h before the main testing session, and from eating 2 h before the session. Water was available throughout the testing session.

### 2.3. Baseline and Post Strength Tests

After a 10-min standardised warm-up on an ergocycle (Monark 874E; Monark, Varberg, Sweden) at 100 W with 6 s intermittent sprints in the last 4 min, participants completed a strength test on an isokinetic dynamometer (Biodex system 4; Biodex Medical Systems Inc., Shirley, New York, USA). Participants’ position and dynamometer settings have been described elsewhere [7]. The strength test was performed on both the dominant (leg used to kick a ball) and non-dominant legs in a randomised order and at an angular velocity of 120°·s^−1^. This velocity is characterised by good test–retest reliability for hamstring-to-quadriceps ratios (standard error of measurement: 5.3 [18]). The range of motion was from 0° (full knee extension) to 90° of knee flexion and participants were given verbal encouragement to provide maximal effort throughout the range. The test consisted of one set of five concentric contractions of the quadriceps, followed by one set of five eccentric contraction of the hamstrings.

The following variables were calculated as the average of the three best contractions, similar to the study of Zhang et al. [19]:Concentric peak torque of the quadriceps (Q_con_, N·m)Eccentric peak torque of the hamstrings (H_ecc_, N·m)Peak H_ecc_/Q_con_Angle-specific H_ecc_, Q_con_ and H_ecc_/Q_con_ at 10°, 20°, 30° and 40° from full knee extension. These were calculated as averages between 0°–10°, 11°–20°, 21°–30° and 31°–40° [9].Rate of torque development in the first 50 ms and the first 100 ms (RTD_50_ and RTD_100,_ N·m·s^-1^) for H_ecc_, Q_con_ and H_ecc_/Q_con_, calculated as the ratio between the change in torque and the corresponding change in time in the first 50ms and 100 ms of contraction, respectively. The onset of contraction was defined as a torque value of 1% of the peak torque produced during the same contraction [19]. The time windows of 50 ms and 100 ms were chosen as the best compromise between reliability and ecological validity. Indeed, ACL injuries usually occur in the first 50ms after initial ground contact [20]. However, Mentiplay et al. [21] showed greater reliability of RFD at 100 ms than 50 ms. The same time windows were previously used in similar research [19].Dominant versus non-dominant legs asymmetry for all these variables: = [(dominant-non dominant)/dominant] × 100.

### 2.4. Simulated Football Exercise: The 90-Minute Ball-Sport Endurance and Sprint Test (BEAST_90_)

The BEAST_90_ is shown in Figure 1. It was chosen because it has been shown to be a valid and reliable simulation of the game; with less than 3% error of measurement for heart rate, sprint times and circuit times [22]. Specifically, the test consists of circuits during two 45-min halves, with 15-min rest in between. The circuits include 12 m and 20 m sprints, running at approximately 75% of maximal speed, jogging, decelerating, backwards jogging, kicking a football, walking and side stepping between cones [22]. Between each circuit, participants did three countermovement jumps (CMJ).

The following variables were measured during the BEAST_90_:During each lap, 12- and 20-m sprint times in sec using timing gates (Brower Timing System, Draper, UT, US).Circuit times in sec using handheld stopwatches (Fastime, Leicestershire, UK).CMJ height in cm using an electronic jump mat (Probotics Inc, US).Heart rate (HR, beats.min^−1^) was continuously recorded using Polar V800 heart rate monitors (Warwick, UK).Rate of perceived exertion (RPE) was reported immediately after each circuit using the Borg scale [23].

For all these variables, an average value for the entire test was calculated to represent players’ absolute performance. In addition, averages were also calculated in the first and last 15-min of the BEAST_90_ [22], to compute the percentage (%) differences in all variables between the start and end.

### 2.5. Statistical Analysis

All data are presented as mean ± standard deviation with 95% confidence interval (CI) Statistical analyses were conducted using SPSS statistical software (version 27.0, IBM Corp., Armonk, NY, USA). Initial tests were performed to check the normality of all variables using the Shapiro–Wilk test. Subsequently, a two-way analysis of variance (ANOVA) with repeated measures was performed to assess the effects of time, angle and their interaction on angle-specific isokinetic variables. Bonferroni post-hoc tests were then used for pairwise comparisons. For peak isokinetic values and RTD, Student T-tests were undertaken to evaluate the effects of the simulated football exercise on these. Finally, the link between absolute performance and performance change (%) during the BEAST_90_ was assessed by a Pearson correlation coefficient. A *p*-value < 0.05 was considered significant. Effect sizes were calculated as Partial Eta Squared $\left(\eta_{p}^{2}\right)$ for the ANOVA and interpreted as no effect (0–0.05), minimum effect (0.05–0.26), and strong effect (0.26–0.64), while Cohen’s *d* represented the effect size for post-hoc tests, and were interpreted as small (>0.2), medium (>0.5) and large (>0.8), [24].

## 3. Results

Data from the BEAST_90_ are presented in Table 1.

### 3.1. Peak Isokinetic Data

Significant decreases were observed in the post-compared to pre-BEAST_90_ values for H_ecc_ in the dominant (−15.1%, t (32) = 6.432, *p* < 0.001) and the non-dominant (−15.5%, t (32) = 4.817, *p* < 0.001) legs (Table 2). The football-simulation exercise also resulted in significant decreases in Q_con_ in the dominant leg only (−7.3%, t (32) = 4.515, *p* < 0.001) and in H_ecc_/Q_con_ in the dominant (−8.8%, t (32) = 1.205, *p* = 0.007) and non-dominant (−12.9%, t (32) = 2.240, *p* < 0.039) legs (Table 2).

### 3.2. Angle-Specific Isokinetic Data

The statistical analysis on H_ecc_ in the dominant leg showed significant effects of time (F (1) = 18.693, *p* < 0.001, η_p_^2^ = 0.369) and interaction between time and angle (F (3) = 3.161, *p* = 0.028, η_p_^2^ = 0.090, Table 3). However, no significant effect of angle were observed on this variable (F (3) = 1.090, *p* = 0.357, η_p_^2^ = 0.033). Post-hoc analyses revealed significant decreases in H_ecc_ between pre- and post-BEAST_90_ at 10° (−22.2%, *p* < 0.001), 20° (−18.1%, *p* < 0.001), 30° (−18.1%, *p* < 0.001) and 40° (−14.1%, *p* < 0.001). In the non-dominant leg, there was a significant effect of time only (F (1) = 5.139, *p* = 0.030, η_p_^2^ = 0.131). Post-hoc analyses revealed significant decreases between pre- and post-BEAST_90_ data ranging from 10.6% to 15.9% (*p* values ranging from 0.028 to 0.037).

The Q_con_ results of the dominant leg showed significant effects of time (F (1) = 8.674, *p* = 0.006, η_p_
^2^= 0.219), angle (F (3) = 427.0, *p* < 0.001, η_p_^2^ = 0.932) and interaction between these factors (F (3) = 15.48, *p* < 0.001, η_p_^2^ = 0.333, Table 3). Post-hoc analyses revealed significant decreases in Q_con_ between pre- and post-BEAST_90_ at 20° (−10.2%, *p* = 0.033), 30° (−11.5%, *p* < 0.001) and 40° (−12.2%, *p* < 0.001). No significant effect of time (F (1) = 0.009, *p* = 0.927, η_p_^2^ = 0.001) or time*angle interaction (F (3) = 0.808, *p* = 0.492, η_p_^2^ = 0.024) were found regarding Q_con_ in the non-dominant leg. In contrast, a significant effect of angle was observed on this variable (F (3) = 48.793, *p* < 0.001, η_p_^2^ = 0.597).

Significant effects of angle (F (3) = 41.720, *p* < 0.001, η_p_^2^ = 0.574) and interaction between time and angle (F (3) = 3.328, *p* = 0.023, η_p_^2^ = 0.097) were observed on H_ecc_/Q_con_ in the dominant leg, with no significant effect of time (F (1) = 2.587, *p* = 0.118, η_p_^2^ = 0.077, Table 3). Post-hoc analyses showed a significant decrease between pre- and post-BEAST_90_ at 10° only (−15.5%, *p* = 0.041). In the non-dominant leg, the results revealed significant effects of time (F (1) = 8.716, *p* = 0.006, η_p_^2^ = 0.209), angle (F (3) = 36.462, *p* < 0.001, η_p_^2^ = 0.525) and time*angle interaction (F (3) = 7.093, *p* < 0.001, η_p_^2^ = 0.177) on H_ecc_/Q_con_. Post-hoc tests showed that H_ecc_/Q_con_ significantly decreased between pre- and post-BEAST_90_ at 10° (−21.8%, *p* = 0.004), 20° (−17.2%, *p* = 0.013) and 30° (−15.1%, *p* = 0.002).

### 3.3. Rate of Torque Development (RTD_50_ and RTD_100_)

The statistical analysis showed a similar pattern for both measures of RTD (Table 4). Significant decreases were observed in RTD_50_ H_ecc_ (−15.0%, t (32) = 2.189, *p* = 0.018) and RTD_100_ H_ecc_ (−17.0%, t (32) = 2.464, *p* = 0.010) between pre- and post-BEAST_90_ in the dominant leg. Similar results were shown in the non-dominant leg (−14.1%, t (32) = 1.815, *p* = 0.040 for RTD_50_ and −14.0%, t (32) = 1.834, *p* = 0.038 for RTD_100_). No significant difference between pre- and post-BEAST_90_ were observed in the dominant leg for RTD_50_ H_ecc_ (t (32) = 0.727, *p* = 0.227) and RTD_100_ H_ecc_ (t (32) = 1.047, *p* = 0.0152) or in the non-dominant leg for RTD_50_ H_ecc_ (t (32) = 1.284, *p* = 0.104) and RTD_100_ H_ecc_ (t (32) = 0.883, *p* = 0.192). Similarly, No significant difference between pre- and post-BEAST_90_ were observed in the dominant leg for RTD_50_ H_ecc_/Q_con_ (t (32) = 0.555, *p* = 0.291) and RTD_100_ H_ecc_ (t (32) = 0.707, *p* = 0.242) or in the non-dominant leg for RTD_50_ H_ecc_/Q_con_ (t (32) = 1.128, *p* = 0.134) and RTD_100_ H_ecc_ (t (32) = 0.234, *p* = 0.114).

### 3.4. Asymmetry

No significant effects of the BEAST90 on asymmetry were shown for all the outcome variables, including peak values, angle-specific value or RTD (p ranging from 0.092 to 0.435).

### 3.5. Correlations between ACL Risk Factors and BEAST_90_ Data

No significant correlation was shown between either absolute performance or performance change during the BEAST_90_ and any of the isokinetic risk factors measured (r ranging from −0.109 to +0.322, *p* < 0.05).

## 4. Discussion

The main findings of the present study showed that fatigue induced by a simulated football exercise led to significant reductions in peak H_ecc_, peak H_ecc_/Q_con_ and rapid rate of torque development for hamstring eccentric contractions in the dominant and non-dominant legs. In addition, while angle-specific H_ecc_ was significantly lower in the post- vs. pre-test at 10°, 20°, 30° and 40° in both legs, significant decreases in angle-specific H_ecc_:Q_con_ were only observed at 10° in the dominant leg, and 10°, 20° and 30° in the non-dominant leg. Regarding our second aim, we found no significant association between the above-mentioned changes and performance or physiological changes occurring during the BEAST_90_.

### 4.1. Peak Isokinetic Data

The significant decrease in peak H_ecc_ and H_ecc_/Q_con_ after a football-specific exercise is in accordance with previous literature on male and female footballers. Indeed, significant decreases in H_ecc_ ranging from −9.4% to −15.6% (compared to −15.1% to −15.5% in the present study, with small to moderate effect sizes) have been previously observed following football matches or football-specific exercise in female footballers of various levels [11,16]. Similar results (mean decreases of −13.9 to −19.6%) have been reported in a systematic review on male football players [13]. The lower eccentric strength of the hamstrings at the end of a football match could be explained by the major involvement of hamstrings eccentric contractions during the various activities undertaken in football, such as sprinting, jumping and decelerating of kicking. In particular, during running-based activities, the hamstrings contract eccentrically to decelerate the forward movements of the thigh and leg and to stabilize the knee in the late part of the swing phase just before the foot makes contact with the ground [25]. The significant decrease in H_ecc_ in the present study, coupled to less reductions in Q_con_ (significant −7.3% decrease in the dominant leg and non significant decrease in the non dominant leg) led to a significant reduction in peak H_ecc_/Q_con_, ranging from 8.8% to 12.9%. These results are very similar to previous data obtained on amateur female footballers [11] and could be explained by the greater proportion of more fatigable type II muscle fibres in the hamstrings compared to the quadriceps [26]. These results go against the assumption of greater resistance to fatigue in females than males [14,15], although no direct conclusion on this could be drawn as we did not test male and female players within the same settings, and comparison with other studies is biased by the various fatigue protocols undertaken.

There are contrasting results in the literature about the association between the variation in peak isokinetic data and performance/physiological changes occurring during simulated or real football matches [27,28,29]. This suggests that vulnerability to injury cannot be predicted from performance changes and hence should be considered as an independent aspect.

While peak H_ecc_/Q_con_ has been widely used in the past 20 years as a risk factor for ACL injury, the functional significance of using peak values is questionable. Indeed, the angle at which the quadriceps concentric peak torque and hamstrings eccentric peak toque occur is different [7]. In addition, ACL injuries usually occur at angles close to full knee extension [8]. Hence, measuring angle-specific H_ecc_/Q_con_ seems more relevant to the understanding of a player’s strength characteristics in the context of injury prevention, in particular at angles close to extension [9].

### 4.2. Angle-Specific Torque

Our results on angle-specific H_ecc_/Q_con_ showed that football-specific fatigue led to significant decreases in H_ecc_/Q_con_ at 10° only in the dominant leg, and at 10°, 20° and 30° in the non-dominant leg. These differences were characterised by moderate effect sizes. De Ste Croix et al. [9] reported similar results on the dominant leg in youth female footballers (decreases of −17.3% and −15.5% at 10°, respectively for de Ste Croix et al. [9] and the present study). Comparable results were also reported on male semi-professional players (−21.8% decrease in H_ecc_/Q_con_ at 10° [7]), and recreationally active men and women (greater decreases in torque at 15° vs. 30° and 45° [30]). In the present study, this lower H_ecc_/Q_con_ at 10° results from a reduced H_ecc_ at this angle (−22.2%) while Q_con_ at 10° was unchanged. Previous studies also observed significant decreases in H_ecc_ after football-specific fatigue, as well as a shift in the angle of peak torque for H_ecc_ towards less extended knee positions [7,31], suggesting that the hamstrings operates on the descending limb of the length-tension curve [32]. While there is no conclusive evidence to explain such as shift, shorter muscle lengths have been associated with greater susceptibility to damage to the hamstrings, and therefore greater risk of hamstring injury [31]. A decrease in H_ecc_/Q_con_ with fatigue when the leg is close to extension could also put the ACL at a greater risk of injury towards the end of a football match, as it reflects a reduced capacity of the hamstrings to counteract powerful quadriceps contraction at these angles, resulting in increased anterior tibial translation. Indeed, it has been shown that ACL injuries often occur with the leg near full extension [8], and that football-specific fatigue led to significant reductions in maximal hip flexion and knee extension angles (due to shorter hamstrings), predisposing the hamstrings and ACL to injuries [33]. These findings are particularly important for female footballers, as many actions in football are performed with the knee at extended positions and females are known to perform landing task with the knee less flexed than males [34]. 

Our results at other angles close to knee extension are in accordance with those from El Ashker, Allardyce and Carson [30], showing significant decreases in hamstrings eccentric torque at angles of 45°, 30° and 15° in recreationally active men and women. In contrast, de Ste Croix et al. [9] showed a significant increase in the torque produced at 30° after fatigue in U17 female footballers, and no significant change at 20°. Various factors could explain these differences. First, de Ste Croix’s participants were much younger than our participants were, and although characterised as post-pubertal, might have still been going through neuromuscular changes linked to puberty [35]. In addition, our players were amateur and were only training twice a week at the time of the study, in contrast with the heavier training load experienced by participants in de Ste Croix’s study. It is well-known that football-specific training leads to quadriceps dominance [36], and hence more trained players might experience less effects of fatigue in this muscle group, hence increasing H_ecc_/Q_con_. All together, these findings suggest that neuromuscular adaptations linked to age, sport specificity or training status might have an influence on the effects of fatigue on angle-specific H_ecc_/Q_con_ and susceptibility to ACL injuries, although more research is needed to investigate these underlying mechanisms.

In the non-dominant leg, we observed significant decreases in H_ecc_ at all angles (from 10.6% to 15.9%), while Q_con_ was not significantly affected by fatigue. Consequently, H_ecc_/Q_con_ was significantly lower after the simulated football exercise at 10° (−21.8%), 20° (−17.2%) and 30° (15.1%), suggesting that the capacity of the hamstring to stabilise the knee at these angles is more affected in the non-dominant than the dominant leg. To our knowledge, our study is the first to assess the effects of fatigue on the torque-angle relationship in both legs, and hence our results cannot be compared with the literature. However, previous isokinetic studies showed a gender difference in the effects of fatigue on peak H_ecc_/Q_con_ values. Significant decreases in H_ecc_/Q_con_ after simulated football exercise were reported in both legs in female players [11], while only the dominant leg H_ecc_/Q_con_ was affected by fatigue in male players [27,33]. These results and the strength profile observed in the present study with fatigue seems to parallel the patterns of ACL injuries reported in epidemiological studies, with male footballers characterised by more ACL injuries in the dominant leg, while no significant differences between legs or a tendency for more ACL injuries in the non-dominant leg were reported in females [37,38]. The greater vulnerability to ACL injuries of the non-dominant leg of our female players could be explained by the role of the non-dominant leg to stabilise the body and absorb shocks during football-specific activities, and in particular, the kicks or jumps of our fatigue protocol [39]. Interestingly, Nakahira et al. [40] showed that during single-leg drop vertical jumps, the knee valgus angle of female footballers at initial contact was greater in the non-dominant compared to the dominant leg. They suggested that these differences might reflect a reduced control of the muscles acting on the knee in the frontal plane due to changes in the contraction patterns of the hamstrings and gastrocnemius muscles acting as knee adductors and abductors of the knee [41]. Within this context, Gehring, Melnyk and Gollhofer [42] observed that fatigue induced a delayed activation of the lateral hamstrings muscle in female players compared with male players in the preparatory phase of landing. However, these studies were not undertaken on footballers and fatigue was induced by non-functional exercises, which highlights the need for further studies in this area.

### 4.3. Rate of Torque Development

In addition to the level of force that can be produced by a player at various knee angles, the rate at which this force is produced seems to be a crucial parameter to consider for both football performance and injury prevention [43]. Indeed, while peak force is usually reached after around 500 milliseconds from the start of contraction, specific football movements (i.e., accelerating, kicking, sprinting, etc.) occur with contraction times lower than 250 ms, in particular with a crucial influence of the rapid strength capacity of the hamstrings within the first 100 ms on acceleration performance of football players [44]. The time available to stabilize the knee in sporting situations where athletes rapidly make contact with the ground is even less (lower than 50 ms [20]) and coincides with the time reported for most non-contact ACL injuries (17 to 50 ms after initial ground contact) [20]. Within this context, Zebis et al. [43] introduced the concept of early RFD H/Q strength ratio during isometric contractions to characterise the capacity of the hamstrings to quickly react to rapid quadriceps contractions. Our results showed significant decreases in RTD_50_ and RTD_100_ for H_ecc_ in both legs (from −14.1% to −17.0%), while no significant changes were observed on either measures of RTD for Q_con_ or H_ecc_/Q_con_. To our knowledge, our study is the first to investigate the effects of fatigue on RTD in female football players. Our results on RFD H_ecc_ are in accordance with those of Zhang et al. [19] in male professional football players (-21.4% decreased RTD H_ecc_ in the first 100 ms). Similar results were also observed on the effects of fatigue on RFD using isometric contractions (−16.0% decrease in RFD in the first 50 ms and from −9% to −24% decreases in RFD in the fisrt 200 ms [28,29]). All together, these results indicate that the capacity of the hamstring to quickly contract to stabilize the leg during tasks such as landing is reduced with fatigue, which could increase the risk of ACL injuries.

Contrasting results were reported on the effect of fatigue on RFD or RTD for the quadriceps or H/Q, with some authors showing, like us, no significant effects of fatigue on these parameters [28], while others showed significant decreases in RTD HQ in the first 100 ms of contraction. The study of Zhang et al. [19] could offer some elements of explanation for these findings, as they concomitantly investigated the effects of fatigue on RTD and electromyographic (EMG) activity of some quadricep (vastus medialis and lateralis) and all hamstring muscles during concentric and eccentric contractions of both muscle groups. They found that the power corresponding to the high frequencies of the EMG signal was significantly decreased in a fatigued state for the quadriceps muscles contracting concentrically; however, this reduction was consistent throughout the contraction. In contrast, during, they reported a reduction in the EMG frequency of the biceps femoris that specifically occurred during the first 30% of the contraction. This suggests that football-specific fatigue could induce specific alterations of the recruitment and firing frequency of the motor units of the hamstrings during the early part of the contraction, resulting in a lower RTD. While these observations could partly account for our results, similar studies are needed on female players, because of gender differences in neuromuscular control. It should also be acknowledged that RTD is influenced by many other factors, such as tissue stiffness, muscle typology, cross-bridge kinetics and neural drive [45].

While all the above-mentioned results may have some implications for the prevention of ACL injuries, there are anatomical and hormonal factors that we did not account for, which could also influence the rate of ACL injuries. First, the wider hips of females increase the valgus stress placed on the knee structures and makes them more susceptible to ACL injuries [2,46]. Second, the lower levels of testosterone and greater levels of estrogen in females lead to reduced muscular support around the knee, and greater laxity of the ACL ligament, respectively, compared to men [35].

The main limitations of the present study include the relatively small sample size used. Thus, this study could be considered as a pilot study. Furthermore, using a football-simulated exercise, even if validated, does not exactly reproduce the intensity of a competitive match. In addition, while our more novel risk factors (angle-specific torque and RTD) have been investigated in several studies [9,19] they need to be evidenced as risk factors in prospective studies. Finally, we could not measure menstrual cycle phases for our players, which could have affected our results due to the influence of estrogen levels on ligament laxity.

## 5. Conclusions

Our results showed significant reductions in peak H_ecc_, peak H_ecc_/Q_con_ and rapid rate of torque development for hamstring eccentric contractions in the dominant and non-dominant legs following a simulated-football exercise. In addition, while angle-specific H_ecc_ was significantly lower in the post- vs. pre-test at 10°, 20°, 30° and 40° in both legs, significant decreases in angle-specific H_ecc_:Q_con_ were only observed at 10° in the dominant leg, and 10°, 20° and 30° in the non-dominant leg. These results suggest that the capacity of the hamstring to stabilise the leg is reduced with fatigue, and possibly to a greater extent in the non-dominant leg. Our findings are different from those obtained regarding male footballers and are important to consider in a perspective of ACL injury prevention in female footballers. It is important to note that these results are only applicable to populations similar to ours (female amateur players training twice weekly), as greater training loads could lead to different effects. Further studies in this area could also investigate the effects of fatigue induced by various workload durations (for example, after 45 min, which would represent one half of a match).

## Figures and Tables

**Figure 1 biology-12-00124-f001:**
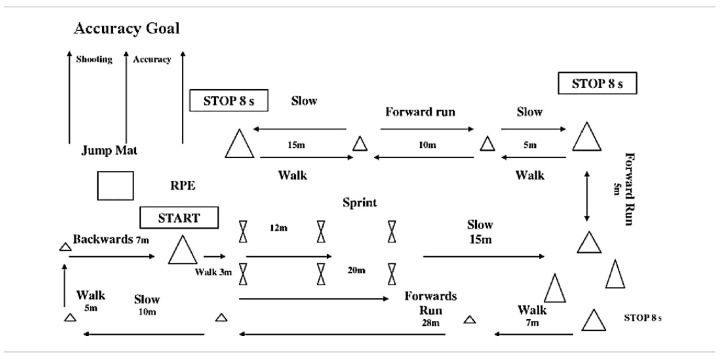
Design of the 90-min ball-sport endurance and sprint test (BEAST_90_, from Williams, Abt and Kilding [22]).

**Table 1 biology-12-00124-t001:** Sprint (12-m and 20-m) times, countermovement jump (CMJ) height, circuit time, ratings of perceived exertion (RPE) and heart rate (HR) averaged different time intervals during the simulated football exercise (BEAST_90_).

	12-m (s)	20-m (s)	Circuit (s)	CMJ (cm)	RPE	HR (%HR_max_)
0–90 min	2.59 ± 0.23	4.29 ± 0.51	202 ± 11	14.2 ± 1.7	14 ± 2	84.6 ± 6.6
0–15 min	2.49 ± 0.21	4.09 ± 0.51	203 ± 13	14.7 ± 1.9	12 ± 2	82.4 ± 5.9
75–90 min	2.62 ± 0.27	4.29 ± 0.51	201 ± 9	13.8 ± 1.5	15 ± 1	85.4 ±7.4
% change	+5.2%	+4.9%	−1.0%	−6.1%	+25%	+3.6%

**Table 2 biology-12-00124-t002:** Peak isokinetic data before (PRE) and after (POST) the simulated football exercise (BEAST_90_).

Variables	PRE Torque (N·m)	POST Torque (N·m)	95% CI for the Difference	Cohen’s *d*
**H_ecc_ D**	126 ± 53	107 ± 47 *	13 to 25	0.38
**H_ecc_ ND**	124 ± 51	105 ± 45 *	11 to 28	0.40
**Q_con_ D**	159 ± 46	147 ± 45 *	6 to 17	0.25
**Q_con_ ND**	147 ± 45	143 ± 48	−3 to 12	0.09
**H_ecc_/Q_con_ D**	0.80 ± 0.18	0.73 ± 0.17 *	0.02 to 0.12	0.40
**H_ecc_/Q_con_ ND**	0.85 ± 0.17	0.74 ± 0.21 *	0.01 to 0.20	0.56

* Significantly different from pre-test values, *p* < 0.05. 95% CI: 95% confidence interval, Q_con_: quadriceps concentric contractions, H_ecc_: hamstring eccentric contractions, D: dominant leg, ND: non-dominant (ND) leg.

**Table 3 biology-12-00124-t003:** Angle-specific isokinetic data before (PRE) and after (POST) the simulated football exercise (BEAST_90_).

Variables		PRE Torque (N·m)	POST Torque (N·m)	95% CI for the Difference	Cohen’s *d*
**H_ecc_ D**	**10**	72 ± 34	56 ± 27 *	−24 to −7	0.52
	**20**	72 ± 31	59 ± 25 *	−20 to −6	0.46
	**30**	72 ± 28	59 ± 24 *	−18 to −7	0.50
	**40**	71 ± 25	61 ± 22 *	−15 to −6	0.42
**H_ecc_ ND**	**10**	63 ± 31	53 ± 31 *	−20 to −1	0.32
	**20**	64 ± 29	55 ± 27 *	−17 to −1	0.32
	**30**	65 ± 26	58 ± 25 *	−14 to −1	0.27
	**40**	66 ± 25	59 ± 24 *	−12 to −1	0.29
**Q_con_ D**	**10**	38 ± 13	35 ± 15	−1 to 7	0.21
	**20**	49 ± 15	44 ± 17 *	1 to 9	0.31
	**30**	61 ± 17	54±18*	3 to 11	0.40
	**40**	74 ± 19	65 ± 21 *	4 to 13	0.45
**Q_con_ ND**	**10**	31 ± 13	34 ± 11	−6 to 1	−0.25
	**20**	49 ± 22	45 ± 13	−11 to 18	0.22
	**30**	55 ± 20	59 ± 16	−6 to 1	−0.22
	**40**	68 ± 22	67 ± 21	−3 to 7	0.05
**H_ecc_/Q_con_ D**	**10**	2.13 ± 1.23	1.80 ± 1.19 *	−0.69 to −0.04	0.27
	**20**	1.60 ± 0.77	1.44 ± 0.78	−0.37 to 0.04	0.21
	**30**	1.23 ± 0.48	1.15 ± 0.53	−0.21 to 0.05	0.16
	**40**	1.00 ± 0.33	0.98 ± 0.37	−0.12 to 0.08	0.06
**H_ecc_/Q_con_ ND**	**10**	2.11 ± 1.02	1.65 ± 1.19 *	6 to 17	0.42
	**20**	1.57 ± 0.71	1.30 ± 0.75 *	−3 to 12	0.37
	**30**	1.19 ± 0.37	1.01 ± 0.42 *	0.02 to 0.12	0.45
	**40**	0.97 ± 0.27	0.91 ± 0.33	0.01 to 0.20	0.20

* Significantly different from pre-test values, *p* < 0.05. 95% CI: 95% confidence interval, Q_con_: quadriceps concentric contractions, H_ecc_: hamstring eccentric contractions, D: dominant leg, ND: non-dominant (ND) leg.

**Table 4 biology-12-00124-t004:** Rate of torque development in the first 50ms (RTD_50_) and100 ms (RTD_100_) before (PRE) and after (POST) the simulated football exercise (BEAST_90_).

Variables	PRE RTD (N·m.s^−1^)	POST RTD (N·m.s^−1^)	95% CI for the Difference	Cohen’s d
**RTD_50_ H_ecc_ D**	312 ± 130	265 ± 89 *	3 to 92	0.39
**RTD_100_ H_ecc_ D**	347 ± 144	288 ± 97 *	10 to 108	0.44
**RTD_50_ H_ecc_ ND**	319 ± 148	274 ± 100 *	−6 to 94	0.33
**RTD_100_ H_ecc_ ND**	350 ± 163	301 ± 110 *	−6 to 108	0.33
**RTD_50_ Q_con_ D**	515 ± 197	502 ± 200	−23 to 49	0.13
**RTD_100_ Q_con_ D**	591 ± 194	570 ± 198	−20 to 62	0.19
**RTD_50_ Q_con_ ND**	516 ± 208	482 ± 170	−20 to 90	0.23
**RTD_100_ Q_con_ ND**	580 ± 204	553 ± 164	−35 to 89	0.16
**RTD_50_ H_ecc_/Q_con_ D**	0.66 ± 0.32	0.63 ± 0.44	−0.07 to 0.12	0.10
**RTD_100_ H_ecc_/Q_con_ D**	0.66 ± 0.30	0.62 ± 0.43	−0.06 to 0.13	0.13
**RTD_50_ H_ecc_/Q_con_ ND**	0.73 ± 0.75	0.59 ± 0.26	−0.11 to 0.38	0.21
**RTD_100_ H_ecc_/Q_con_ ND**	0.74 ± 0.76	0.58 ± 0.27	−0.10 to 0.40	0.23

* Significantly different from pre-test values, *p* < 0.05. 95% CI: 95% confidence interval, Q_con_: quadriceps concentric contractions, H_ecc_: hamstring eccentric contractions, D: dominant leg, ND: non-dominant (ND) leg.

## Data Availability

Data is unavailable.

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
