# Peer review of "Effects of a Football Simulated Exercise on Injury Risk Factors for Anterior Cruciate Ligament (ACL) Injury in Amateur Female Players"

_biology, 2023, doi:10.3390/biology12010124_

Round 1

Reviewer 1 Report

Dear authors thank you very much for the submission.

Some comments to consider:

1)      Thirty-three female football players volunteered to take part; provide details how you recruited them.  

2)      Based on eyeball scans this sample size is underpowered statistically at alpha 0.05 beta 0.2 and power 80% based on their data clearly all Cohens es <0.5 thus I urge authors to first call this a pilot/preliminary finding and second to provide formal and convincing sample size instead of referring readers to de Ste Croix et al. (2018)’s paper. Provide assumptions us used for G*power and provide used equations.

3)      Table 4 Cohens d need to be x.xx

4)      The authors demonstrated that, after a simulated football workout, hamstring eccentric contractions in the dominant and non-dominant legs decreased significantly in peak Hecc, peak Hecc/Qcon, and the pace at which torque developed. This was not discussed adequately for injuries/injuries prevention specially with the findings regarding angle-specific torque.

Because of the broader female pelvis, the thigh bone, tibia, and femur function differently than in men and this explain larger acl in women.

Women's knee muscles are less than men's in the knee region.

Hormones not mentioned at all in discussion. Women have less testosterone than men, which is a crucial hormone for building more muscle mass. Compared to men, women also have a lot more estrogen. This hormone changes during a woman's menstrual cycle and is crucial for bone formation. Additionally, estrogen may increase the laxity (looseness) of tendons and ligaments, making women more vulnerable to injury.

Menstrual cycle (lack of its measure) is limitation.

Reviewer 2 Report

ACL injury is one of the most common and most the serious injuries in sports. That is reason enough to pay attention to this topic. Also, it is well-known that this injury is much more common in women than in men.

Introduction section support very good background for this topic. The aims are supported by an adequate and precise description of previous papers. The study is quite intriguing with the appropriate and thorough methods. Statistical analysis was conducted in a very precise manner, with results well presented. Manuscript is rather well written, with important and highlighted practical implications.

There is only one suggestion to the authors, as an addition to the Conclusion section. The authors should state any shortcomings of the study that they are aware of (e.g. subjects who train only 2 times a week is a significant disadvantage and it can be assumed that with more training results could be different). Also, it would be interesting to see the results after different training periods. The match lasts 90 minutes and a lower level of performance is to be expected after that period. However, what are the effects after, for example, 45 minutes would be perhaps even more significant because we might determine the degree of risk in different periods of the match or training. That could be, an addition for future investigations part (or something else important for authors).
